# Transcriptome profiling of barley and tomato shoot and root meristems unravels physiological variations underlying photoperiodic sensitivity

Michael Schneider[1¤a], Lucia Vedder[2], Benedict Chijioke Oyiga[1], Boby Mathew[1], Heiko Schoof[2], Jens Léon[1], Ali Ahmad Naz[1¤b]*

1 University of Bonn, Institute of Crop Science and Resource Conservation, Plant Breeding, Bonn, Germany, 2 University of Bonn, Institute of Crop Science and Resource Conservation, Crop Bioinformatics, Bonn, Germany

¤a Current address: Institute for Quantitative Genetics and Genomics of Plants, Heinrich Heine University, Düsseldorf, Germany
¤b Current address: Faculty of Agriculture and Landscape Architecture, Plant Breeding University of Osnabrück, Osnabrück, Germany
* a.naz@hs-osnabrueck.de

**Data Availability Statement:** All relevant data are within the paper and its Supporting Information

## Abstract

The average sowing date of crops in temperate climate zones has been shifted forwards by several days, resulting in a changed photoperiod regime at the emergence stage. In the present study, we performed a global transcriptome profiling of plant development genes in the seedling stage of root and shoot apical meristems of a photoperiod-sensitive species (barley) and a photoperiod insensitive species (tomato) in short-day conditions (8h). Variant expression indicated differences in physiological development under this short day-length regime between species and tissues. The barley tissue transcriptome revealed reduced differentiation compared to tomato. In addition, decreased photosynthetic activity was observed in barley transcriptome and leaf chlorophyll content under 8h conditions, indicating a slower physiological development of shoot meristems than in tomatoes. The photomorphogenesis controlling cryptochrome gene *cry1*, with an effect on physiological differentiation, showed an underexpression in barley compared to tomato shoot meristems. This might lead to a cascade of suspended sink-source activities, which ultimately delay organ development and differentiation in barley shoot meristems under short photoperiods.

## 1. Introduction

Flowering plants are divided into two major classes—monocotyledons and dicotyledons. Significant diversification of these plants endured about 200 million years ago [1]. In spite of a protracted evolutionary divergence, most cultivated crops are a member of these major categories. Barley and tomato are genomic models for crops representing monocots and dicots, respectively. In addition, these species reveal characteristic differences in their development and growth habits, especially in the root and shoot forms. Thus, genomic dissection of this

files. Raw sequencing data are available in the NCBI (accession SUB1195739).

**Funding:** The authors received no specific funding for this work.

**Competing interests:** The authors have declared that no competing interests exist.

**Abbreviations:** MACE, massive analysis of cDNA Ends sequencing; GO, gene ontology; LFC, log fold change; DEG, differentially expressed gene; EEG, equally expressed gene.

variation provides an opportunity to address critical biological questions behind the evolutionary and developmental divergence.

Roots are programmed in the root apical meristem and part of the elongation zone where the lateral root arises [2]. Shoots develop in the shoot apical meristem and its peripheral location, where leaf primordial arises successively. A major factor determining the development rate is the phyllochron, which ultimately regulates branching [3]. Besides the temperature as a significant factor determining the phyllochron [4], the photomorphogenesis is light-mediated [5]. In Barley and Tomato, three orthologous cryptochrome-mediated light response genes were characterized and described (*Cry1a*, *Cry1b*, *Cry2*) [5]. All of these have the function of photoreceptors in common [6], where *Cry1a* was described to significantly influence the partitioning of photoassimilates between roots and shoots in tomato [7]. This underlines that although root and shoot develop and grow at different spots, active communication and exchange between both organs determines specific plant architecture [8,9].

Many photoperiod-regulated genes in barley have been described to affect development. The effect of photoperiod-sensitive alleles on the phyllochron and the prior-anthesis developmental phases in barley has been described before [10]. This also highlights the high relevance of fast canopy and root establishment in Mediterranean climates. Little impact of photoperiod sensitive allele Ppd-H1 compared to reduced sensitive allele ppd-H1 regarding the pre-awn primordium stage developmental time variation was found [11].

Contrasting to barley, cultivated tomato is characterized by a day-length neutral growth habit [12]. So far, the effect of photoperiod on the generative development and yield formation in crops has been illustrated [13,14]. Still, little research was performed concerning vegetative development in early growth stages. The growing season extended by up to 20 days in the past decades [15,16], but little gains in biomass production were reported for photoperiod-sensitive species [17,18]. The missing adaptation to these changed growth conditions might cause yield reductions or counteract yield increases in new spring-type varieties. Especially with more frequent drought events observed, unproductive growth habits determined by the photoperiod are undesired [10]. By comparing a photoperiod-dependent and independent species, developmental variations in root and shoot tissues should expose the effect of photoperiodic regulation in a short day length regime of 8h.

## 2. Materials and methods

### Plant material and experimental design for transcriptome analysis

Spring barley cultivar Scarlett and tomato cultivar Moneymaker were used as genotypes in the presented study. Seeds were pre-germinated and sown in soil in 96-cell plant growing trays. Plants were grown inside a growth chamber at 22˚C for 8 hours light and 18˚C for 16 hours night regime for ten days at 60% humidity. Root and shoot apices were harvested the following day, pooling 50 individuals of the same genotype in each of the three biological replicates. Apices were dissected and separated under a microscope. The soil was removed carefully for the root apices by washing these in a petri dish in freshwater. Seven millimeters of primary roots containing the apical meristem and elongation zone of barley and tomato were harvested. The absence of root hairs determined the root elongation zone. Likewise, three biological replicates were harvested independently in each species. Barley vegetative shoot apices comprising apical meristem and emerging leaf primordia were dissected, and 50 shoot apices were pooled in each of the three biological replicates. Similarly, 50 tomato vegetative shoot apices were collected, comprising shoot apical meristem and emerging leaf primordial. Samples were harvested at similar time points on the same day in the laboratory and immediately frozen in liquid nitrogen after dissection.

## RNA extraction and Massive Analysis of cDNA Ends (MACE) analysis

MACE-based transcriptome analysis was performed by GeneXpro GmbH (Frankfurt, Germany) [19]. According to the manufacturer's description, the root and shoot tissues were homogenized, and total RNA was extracted for each sample using the INVITEK plant RNA mini kit (INIVTEK, Germany). RNA was fragmented, and polyadenylated mRNA was enriched by poly-A-specific reverse transcription. A specific adapter was ligated to the 5' ends, and the 3' ends were amplified by competitive PCR. MACE sequencing is based on the True-Quant method, which reduces the amount of duplicate transcript sequencing and enables the precise comparison and identification of ultra-low expressed transcripts. Sequencing was performed on the Illumina platform (San Diego, USA).

## Gene expression and function analyses

Transcriptome data were qualitatively adjusted using Trimmomatic SE (version 0.36) [20] with a minimum length of 40 bases and quality filter parameters of 28 for the leading and 17 for the trailing bases linked with a *head crop* of 10 bases. Fragments were aligned with the barley (IBSC_V2) and tomato (SL2.50) reference genome [21] using BWA mem (version 0.7.17) [22] applying standard settings. Read filtering was performed strictly, applying a quality filter of >60 using samtools 1.8 [23] view option. Duplicates as residuals from the PCR step in sequencing were disregarded due to the low impact in expression analysis [24] and the True-Quant technique. Fragments were matched to the genes by the tool *featurecounts* of the subread software package (version 1.6.2) for tomato and barley separately, using the corresponding annotation files for the used reference [25].

Further analyses were performed in the R (3.4.4) [26] and Julia (1.5.1) [27] environments. The expressional and functional analysis methodology is presented in the workflow chart in S1 Fig. The read count normalization and probability values were calculated using Bioconductor package *edgeR* for transcriptome analysis [28,29]. The analysis of both species was performed separately between the root and shoot apical meristem transcriptomes. Probability ($p$) value adjustment was performed by R function *p.adjust*, once by false discovery rate (FDR) and Bonferroni adjustment. Analysis was further performed based on FDR. Replicate testing was done applying a generalized linear mixed model based on a negative binomial distribution. Differentially expressed genes (DEG) were selected based on FDR values of $p < 0.01$ and a log fold change (LFC) bigger 3. These were used to run a gene ontology (GO) enrichment by AgriGoV2 [30] with default settings. Additionally, the GO terms were compared based on the expression level of the genes. DEG were associated with corresponding GO terms,

$$expr_k = \frac{\sum CPM_{i_k}}{count(i_k)} \tag{1}$$

where the expression *CPM* of all genes *i*, annotated to the same GO term *k* were summarized to an average gene ontology expression $expr_k$. The expression pattern of the root and shoot group was compared in a generalized linear model (based on a negative binomial distribution),

$$p = glm \begin{bmatrix} a_{CPM_{1_k}} & b_{CPM_{1_k}} \\ \vdots & \vdots \\ a_{CPM_{n_k}} & b_{CPM_{n_k}} \end{bmatrix}, \text{ expression} \sim \text{species} \tag{2}$$

where *p* is the probability derived from the *glm* model, which tests the gene expression *1:n* for

GO term *k* in tissue *a* against tissue *b*. This step identified p values for GO terms based on the expression level. After FDR adjustment, candidate GO terms were selected by applying a cut-off of FDR < 0.01, LFC > 3, and a minimum of five genes per GO.

Subsequently, tomato and barley were directly compared on GO expression level, analog to the previously presented equation. Therefore, the gene expression values of the GO terms were matched for both barley and tomato tissues. The LFC between root and shoot of the same species was calculated and compared to the other species for each GO term. Therefore, the LFC distribution of each GO term was compared between tomato and barley. Differentially expressed GO terms were selected from this set based on an FDR < 0.05 and a gene count > 2.

Furthermore, orthologous genes were identified based on reference proteome sequence level using default settings of OrthoMCL [31]. The minimum cut-off was set to an e-value threshold of $1e^{-5}$ and a length match of at least 50% for the essential all-vs-all BLASTP [32] step. The identified set of genes was used to match genes on a 1:1 base by the read count step onward. The genes were extracted from the raw read counting file. Root and shoot were separated so that barley and tomato were compared on both tissue levels separately by edgeR. The output of this was clustered in significantly DEG (FDR < 0.05, LFC > 3, normalized expression in both species >5) and equally expressed genes (EEG)(FDR > 0.2, mean normalized expression over both species >5). The group of orthologous genes was clustered by principal component analysis. The DEG and EEG group were compared on gene count level in the three sub-categories molecular function, cellular component, and biological process. These orthologous genes should provide a classification of evolutionary conservation patterns. Furthermore, orthologous genes, annotated to one of three selected gene ontologies were examined for their chromosomal identity between the species in a circos plot. EEG and DEG were separated to investigate positions similarities and variations and if these were correlating with genomic positions.

Venn diagrams were prepared using the R package VennDiagram 1.6.20 [33]. Bioconductor packages ComplexHeatmap 2.6.2 [34] and Circlize 0.4.11 [35] were used to create the heat-maps. Correlations were performed by corrplot 0.84 package [36]. GO term bar plots were printed using ggplot2 3.3.2 package. Principal component analysis and plots were created by PCAtools 2.2.0 [37] and complex boxplot with either ggplot2 3.3.2 or ggpubr 0.4.0 packages [38,39]. Finally, circos plots were created with OmicCircos 1.2.0 [40].

### Assessment of photoperiod-induced physiological development

An additional experiment was performed to validate one distinct observation in the transcriptome comparison. Therefore, Scarlett and Moneymaker seeds were pre-germinated, and each plant was placed into a pit soil container of 6x6x8 cm size. Ten replicates of each genotype were placed in two different growth cabinets–defined by an 8 and 16h light regime. Other environmental parameters were adjusted to the same settings as priorly described. On the 10th & 11th days after germination, the relative chlorophyll content was measured by a MultispeQ V2.0 hand-held chlorophyll meter (https://www.photosynq.com/), using protocol *PSII measurements* in the center of the leaf. The oldest and the youngest leaf per plant were measured in the center part on both days, seven hours after the lights were switched on. A t-test was performed to compare the relative chlorophyll content between (I) the two photoperiods for tomato and barley separately and (II) the species at the same photoperiod.

## 3. Results

The presented study covered multiple comparison levels to provide a general overview of physiological processes in different tissues and species at the seedling stage under a short-day

photoperiod regime. On the first level, root and shoot meristems within the species were tested against each other to uncover tissue-related gene expression variations. Transcriptome analysis using MACE revealed 7.9 million reads in barley root apices, of which 5.5 million reads were aligned across the barley genome. Among the mapped reads, 4.7 million reads were aligned to barley annotated genes. More transcripts (10.5 million) were found in the barley shoot, of which 5.9 million reads were mapped and aligned with barley annotated genes (S2A Fig). In tomato root and shoot apices, around 12.4 and 7.9 million reads were identified, of which 7.1 and 5.5 million reads were mapped to annotated genes, respectively (S2B Fig). Across tissues and species, the three replicates indicated high similarities (S3 Fig). Pearson correlations of the normalized gene expression between the replicates ranged from 0.97 to 1.00. The calculated p-value between the replications for all genes supports the similarities of replicates observed in the correlation analysis ($r > 0.99$).

## Barley tissue comparison

We detected 16,842 of 39,809 genes (42%) to be expressed in both barley tissues (Fig 1A), from which 1,918 were significantly upregulated in the shoot and 2,214 in the root meristem (Fig 1B and 1C). Additionally, 1,085 genes have only been expressed in the root, while 1,533 genes show expression only in shoot tissue (Fig 1A, S1 Table). By performing a gene ontology enrichment based on the gene cluster occurrences in the overexpressed genes, 51 significant ontology

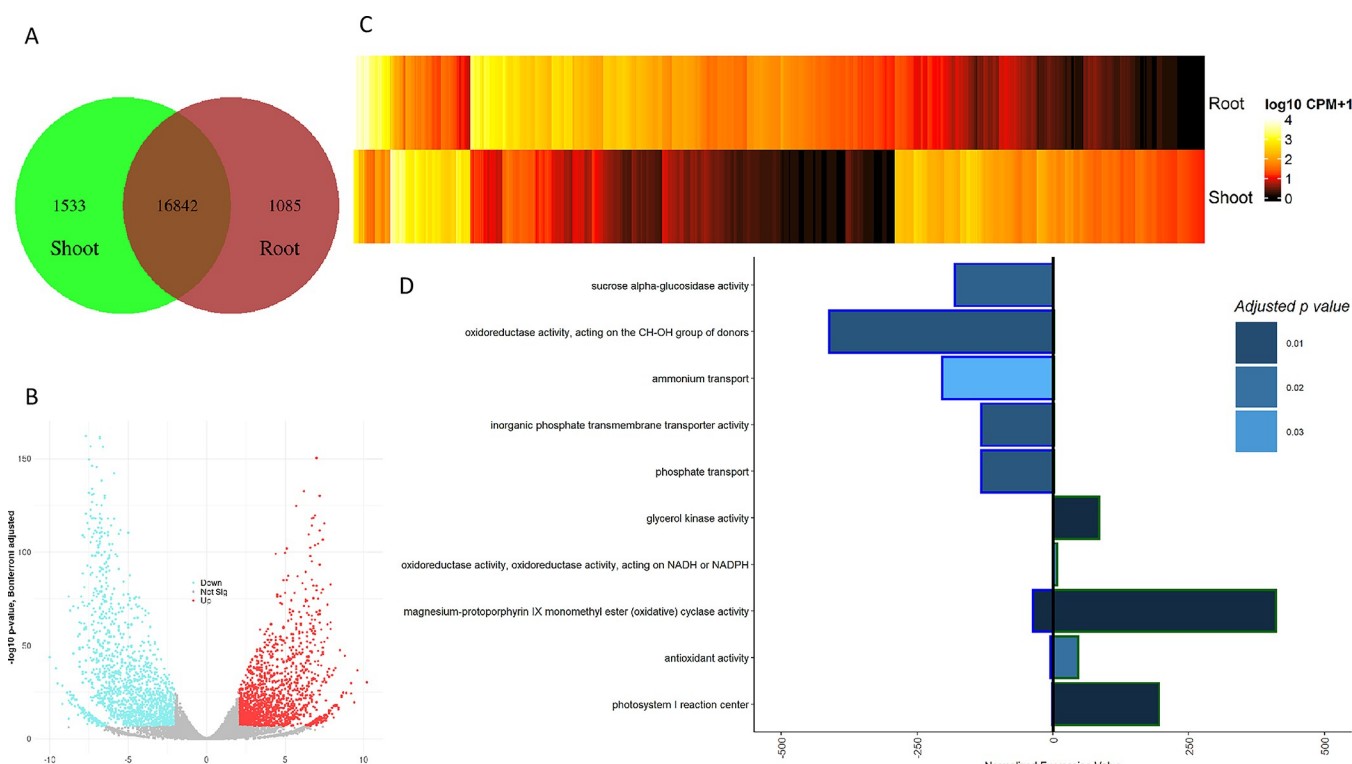

**Fig 1. Overview of root to shoot tissue expression in barley.** A–count of expressed genes in shoot tissue only, both tissues and root tissue only. B–volcano plot of differentially expressed genes, visualizing the Bonferroni adjusted -log10 probability value against the log2 fold change. Blue dots indicate significantly upregulated genes in root tissue; red indicates the same for shoot tissue. C–heatmap of all DEG for the root and shoot tissue. The mean expression value over the three replicates is shown on a log10 transformation. D–differentially expressed gene ontologies. The exterior color of the bar splits root (blue) from the shoot (green), the p-value is indicated by the bar fill. The bars represent the average normalized expression value for the GO terms, based on all genes related to the GO term.

classes were observed (S4 Fig, S2 Table). The most significant GO terms were identified for cell wall organization, oxidation-reduction processes, and heme-binding (p < 0.0001). On a lower but still significant level of probability (p < 0.05), transcription factor activities and metabolic process regulations have been observed to vary between the root and shoot. These observations were further quantified by an expression level analysis of GO terms (S3 Table). Ten significantly different ontologies were detected, where five of these show an up-regulation in root tissue (Fig 1D). Three of these genes are related to the transport of nutrients (ammonium transport, phosphate transport, transmembrane transporter), while one is related to the energy process (sucrose alpha-glucosidase), and the last and strongest is associated with oxidative stress response (oxidoreductase activity). Another family of oxidoreductase genes is significantly upregulated in the shoot meristem, but the absolute expression is much lower than in the root tissue.

Furthermore, two other oxidative response classes were observed (cyclase activity, and anti-oxidation activity). Besides these three stress-responsive ontology classes, the photosystem I reaction center and glycerol kinase activity were overexpressed. Thus, concluding the observations made on the barley tissue comparison level, a sink source pattern can be observed with the additional oxidative stress response.

## Tomato tissue comparison

In tomato, 20,297 of 33,812 genes (60%) were expressed in both tissues, from which 1,537 were upregulated in the shoot and 1,874 upregulated in the root meristem. Additionally, 1,835 genes have shown expression in the root, while 1,362 genes are only expressed in the shoot tissue (Fig 2A–2C, S4 Table). Gene ontology clustering based on the DEG resulted in 63 significant variations between root and shoot tissue (S5 Fig, S5 Table). The most pronounced variation was noticed for stress response, cell wall organization, metabolic processes, transcription factor binding, transporter activity, and photosystem I. By comparing the gene expression level of root and shoot tissue against each other on the GO term level, 27 significantly different classes have been observed (S6 Table). Eleven of these (41%) are overexpressed in the root compared to the shoot tissue, while 16 are overexpressed in the shoot tissue. Most of the root-related GO terms can be classified in the functional groups of transporting (transmembrane transporter), reservoir activity (nutrient reservoir activity, beta-carotene mono oxidase activity, anthocyanin glucosyltransferase), nutrient uptake (Nicotianamine biosynthesis), and elongation (apoplast, gibberellin oxidase). For the shoot, few superordinate classes were identified, which are growth (elongation), photosynthetic activity (protochlorophyllide reductase, photolyase, rubisco, photosystem I and II, chlorophyll-binding, extrinsic to membrane), oxidative response (Flavonoid biosynthesis, oxidoreductase, formamidase activity), developmental activities (indole acid carboxyl transferase) and energy transformation (acetyl-CoA reductase, glyoxylate reductase). The highest overall expression can be reported for photosynthesis-related processes directly linked to photosystems I and II (Fig 2D).

## Comparison of barley and tomato development

While the roots showed equal functional activity between barley and tomato gene ontologies, the shoot tissues varied in photosynthetic activity levels. Remarkably, the oxidoreductase activity (acting on CH-CH group donors) was overexpressed in tomato shoot tissue and barley shoot tissue while lowly expressed in root tissues (Figs 1D & 2D). Compared to the barley GO clustering, 2.7 times more differentially expressed GO terms were observed. The reduced expressional activity in barley tissues resulted in a lower tissue-specific differentiation level (Fig 3A). The principal component analysis revealed variations between species on gene

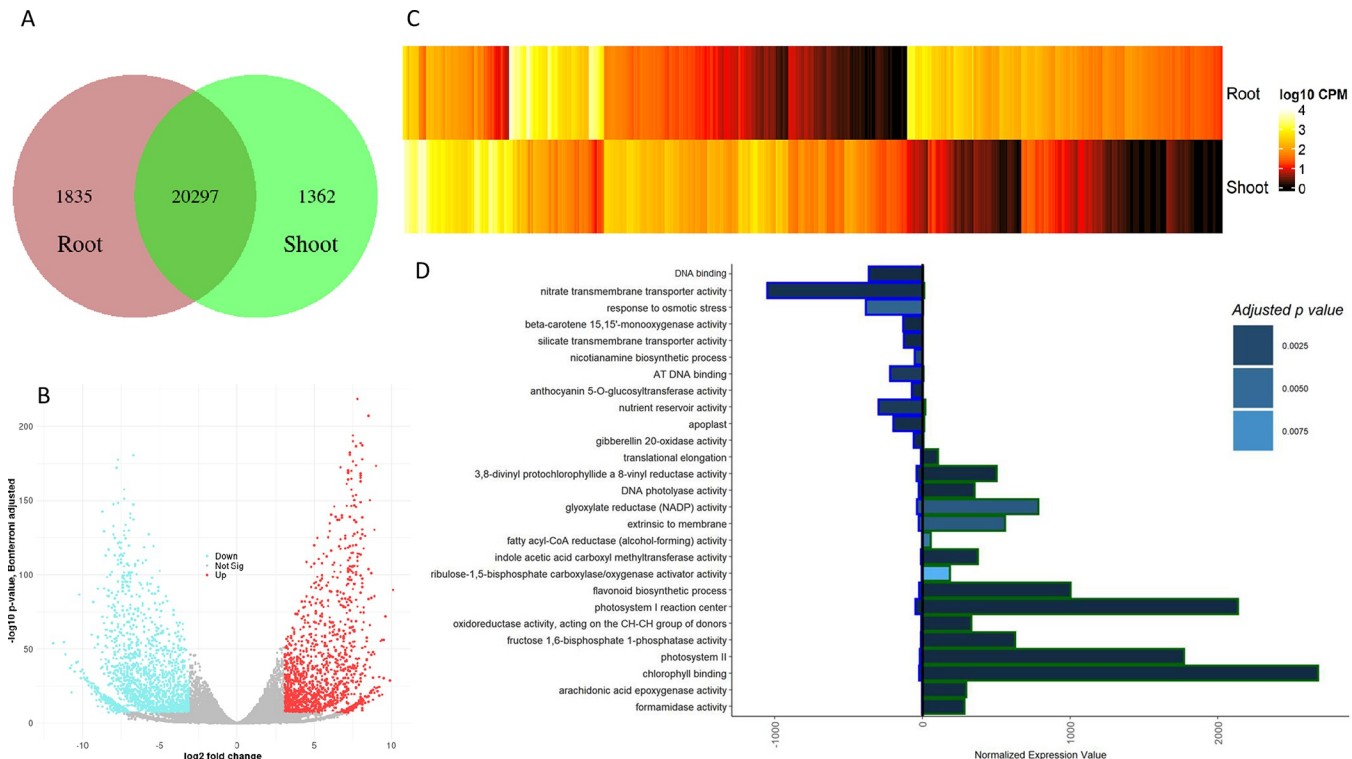

**Fig 2. Overview of root to shoot tissue expression in tomato.** A–count of expressed genes in shoot tissue only, both tissues and root tissue only. B–volcano plot of differentially expressed genes, visualizing the Bonferroni adjusted -log10 probability value against the log2 fold change. Blue dots indicate significantly upregulated genes in root tissue; red indicates the same for shoot tissue. C–heatmap of all DEG for the root and shoot tissue. The mean expression value over the three replicates is shown on a log10 transformation. D–differentially expressed gene ontologies. The exterior color of the bar splits root (blue) from the shoot (green), the fill color indicates the p-value level. The bars represent the average normalized expression value for the GO terms, based on all genes related to the GO term.

ontology expression level, presented by the first component. This first component explains more than 70% of the entire variation. The second component explains 17%, related to the tissue-specific variation. Generally, a more pronounced differentiation of the tomato tissues than the barley tissues was observed. Based on the LFC comparison between barley and tomato, 12 GO terms were identified to be significantly different between the two species (Fig 3B).

Generally, three major groups can be clustered from the direct GO expression comparison. The first group is photosynthetic activity (Photosystem II, Chlorophyll binding) (S6C & S7C Figs). The activity was highest in tomato shoot tissue, while little expression was detected in barley. Two genes were expressed in the barley shoot, while three were in tomato. The average expression level was 100 times higher in tomato than in barley (Tomato 1771; Barley 18 normalized reads). For both species, the Photosystem II reaction center W protein was found to be active (HORVU1Hr1G078140 [5 reads expressed in the shoot meristem], Solyc06g084050 [1720 reads] & Solyc09g065910 [3505 reads]). In contrast, the photosystem I indicated similar expression patterns in barley and tomato (Figs 1D & 2D). The second group was related to stress response on the cellular level, including response to wounding, peroxidase activity, and defense response. While the peroxidase activity differed between the species on $p < 6.5e^{-10}$, the intraspecies tissue variation was $p < 0.001$ (Fig 3B). Fifty-nine genes related to this category were expressed in tomato, showing an average expression level of 943 normalized reads in the root and 124 in the shoot meristem. Contrasting, 256 genes were expressed in barley, with an average expression level of 288 normalized reads in the root and 88 in the shoot meristem. The

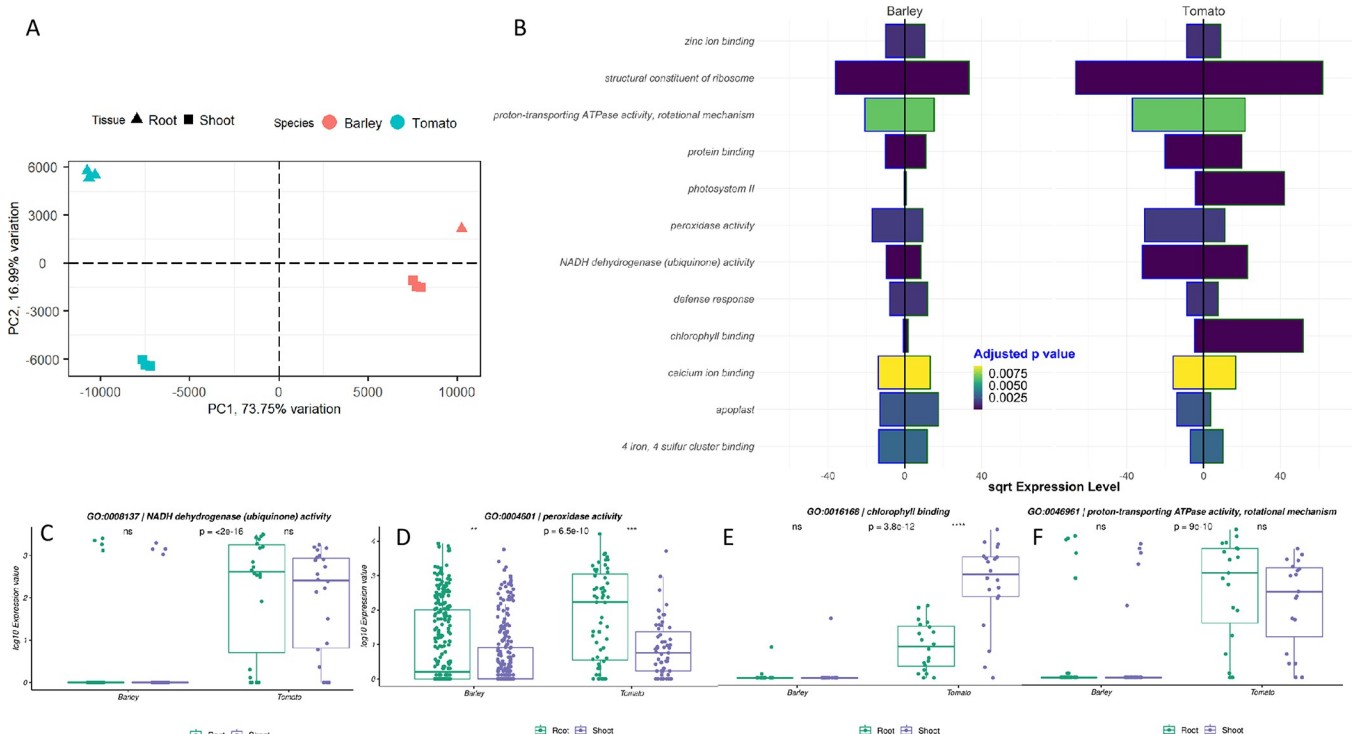

**Fig 3. Gene ontology comparison between barley and tomato.** A -Based on the expression values of genes annotated to a GO term, GO terms were merged between barley and tomato. A principal component analysis of all GO terms matched was performed, comparing all replicates of barley root, barley shoot, tomato root, and tomato shoot tissue to each other. B–the square rooted expression level of barley and tomato tissues for significantly different GO terms, indicated by the fill color of the bars.

highest expressed gene in tomato was the TPX1 peroxidase (Solyc07g052510). Contrasting, the peroxidase superfamily protein (HORVU2Hr1G125110) was in barley (S6B Fig). From these values, we can conclude that the peroxidases are more active in the root than in the shoot meristem in both species. Besides the peroxidase activity, defense response and the response to wounding were discovered for this group. In the category of defense response, we observed an adverse pattern (S7E Fig). While the number of genes was similar (107 tomato genes, 120 barley genes), the average expression of gene ratio was higher in tomato root than shoot meristems (75 to 56 normalized reads). Contrasting, the expression in barley shoot meristems was higher (138) than in root meristems (62). Still, most associated genes in this class were not expressed in either tomato or barley. The highest expressed genes had the same function (Defensin-like protein) in both species (Solyc07g006380, 2,388 expressed reads; HORVU1Hr1G010250, 4,420 expressed reads on average across both tissues).

The response to wounding showed a similar expression pattern in both species (S7I Fig). The expression of all 21 and 30 genes associated with this group was similar in tomato and barley, respectively. However, when considering average expression across all genes, a sildely overexpression in the shoot compared to the root meristems was observed. While a chymotrypsin inhibitor (HORVU1Hr1G004150 [5861 reads on average across both tissues]) was the highest expressed gene in barley, a chymotrypsin inhibitor-2 (Solyc08g080630 [1523]) showed the highest expression of all genes associated with response to wounding. Notably, the gene with the highest expression was observed in barley, while most genes were not expressed in barley meristems.

Interestingly, also chlorophyll-binding indicated variations between the tissues and species. Unsurprisingly, the expression of chlorophyll-binding related genes was higher in shoot than root meristems (S6C Fig). The average expression of 20 genes related to this category was 22 and 2379 normalized reads in tomato root and shoot meristem, respectively. The highest expressed gene was Solyc07g047850, a chlorophyll a-b binding protein. Contrasting, barley tissues showed barely any expression in both tissues. Across 18 genes related to this category. The average expression rate was 0 and 3 normalized reads in root and shoot meristems, respectively. The highest expressed gene was HORVU2Hr1G060480, a photosystem I reaction center subunit (average expression in shoot meristem = 49 reads).

Generally, the expression magnitude for the GO terms was higher in root tissues, with an overall higher expression in tomato. A higher expression level in tomato was also observed for the other two stress-responsive GO terms (S7E & S7I Fig).

Finally, the biggest group was related to respiratory and developmental processes. NADH dehydrogenase, 4 iron 4 sulfur cluster binding, and proton-transporting ATPase activity might be associated with respiratory functions. The ribosomal constitution, protein binding, and calcium-binding appear to be related to developmental processes. In tomoto, 132 genes were found to be related to the calium ion binding category (S7A Fig). The average expression level in root and shoot was not significantly different (253 & 283 normalized reads in root and shoot). The highest expressed gene was Solyc10g081170, a Calmodulin 2 messenger molecule (4590 reads across both tissues). Compared to tomato, about three times more genes (330) were related to this category. The average expression level was lower (190 & 177 for root and shoot meristem), and the highest expressed gene was a calreticulin 1b protein (HOR-VU2Hr1G121990 [5995 reads]). While the tissues did not indicate significant variations in both species, the species themselves significantly diverged from each other (p<0.001).

A similar observation was made for the biggest gene ontology category–protein binding. A total of 2,231 and 3,561 genes were reported for this category in tomato and barley, respectively (S7D Fig). The average expression across both tomato tissues was significantly higher compared to barley (p<0.001; on average, 400 in tomato and 111 in barley). The highest expressed gene in tomato was a meloidogyne-induced giant cell (Solyc01g099770, [24,442 reads across both tissues]). Contrasting to the tomato tissue comparison, barley tissues significantly differed in gene expression (p<0.001).

The category of the structural constituent of ribosomes included 171 genes in tomato and 660 in barley (S7G Fig). In barley, the majority of genes of this category were not expressed. Still, a distinct subset of genes indicated similarly high expression compared to tomato genes. The average expression level of tomato genes was 4183 normalized reads, with the highest expressed gene being a ribosomal protein L3 (Solyc01g104590 [17,050 reads across both tissues]).

Concluding, no significant variation between the tomato tissues was observed for the developmental processes, but a significantly increased expression compared to the barley tissues (S7A, S7D & S7G Fig). The same holds for the respiratory GO terms. No variation between tissues of the same species was observed. Nevertheless, significant overexpression of genes in the tomato tissues was detected compared to the barley tissue (S6A, S6D & S7H Figs).

Subsequently, we wanted to compare the expression of *Cry1* and *Cry2* in both species and tissues. These cryptochromes were reported to mediate the photoperiodic control of flowering, entrainment of the circadian clock, cotyledon opening and expansion, anthocyanin accumulation, and root growth [41]. The *blastn* of Hv-CRY1a/b and Hv-CRY2 sequences, derived from [42], revealed *HORVU6Hr1G049950* (Hv-CRY1) and *HORVU6Hr1G058740* (Hv-CRY2) as single hits. We compared the expression of Hv-CRY1/2 in root and shoot to the expression of orthologous tomato *Cry1* and *Cry2* genes (Fig 4). Locus information of tomato orthologous

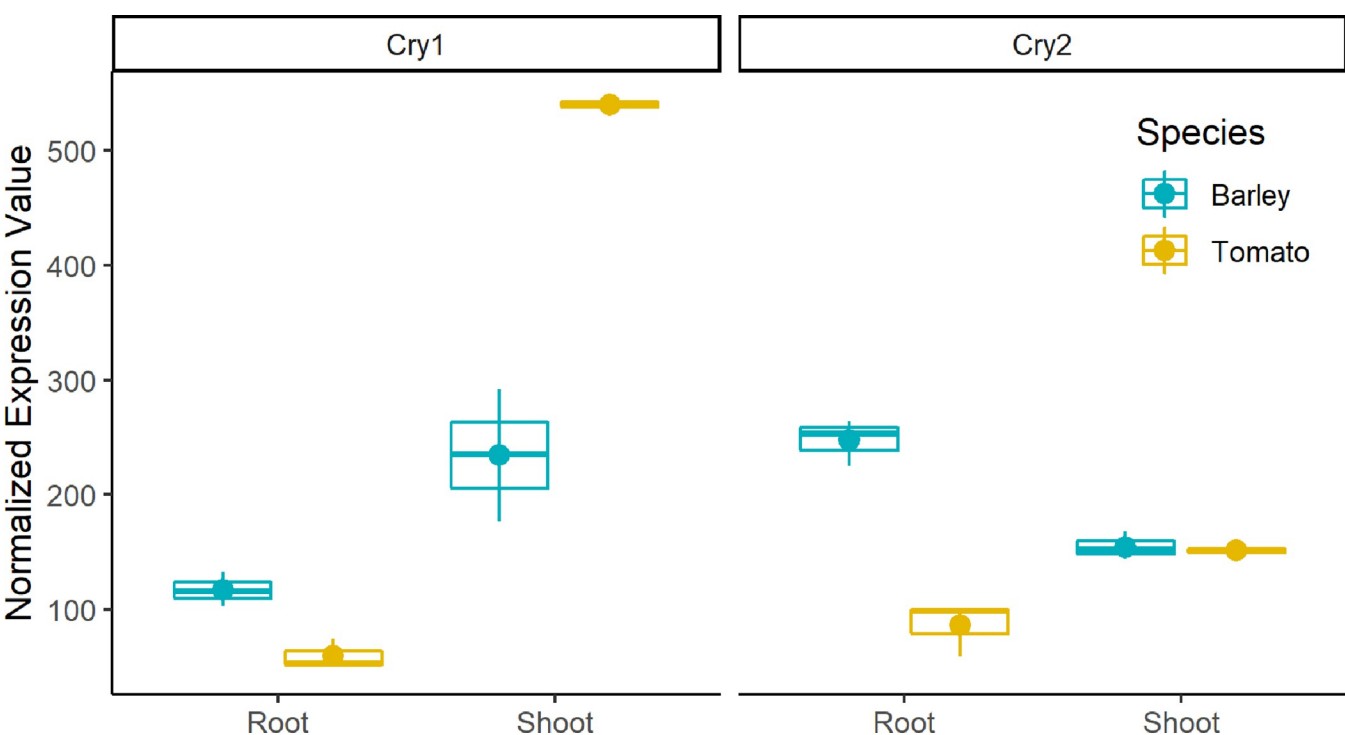

**Fig 4. Normalized expression (TMM) of photomorphogenesis controlling genes *Cry1* and *Cry2*.** The expression is compared between tissues (x-axis) and species (color).

was derived from [41]. While we did not observe variations in the gene expression of *Cry2* in shoot tissues between tomato and barley, *Cry1* revealed 2.3 times higher gene expression in tomato compared to barley shoot. The root tissue expression of *Cry1* and *Cry2* revealed a 1.95 and 2.87 times overexpression in the barley root tissue compared to tomato, respectively.

Finally, we aimed to validate the observations made in terms of divergent photosynthetic activity, mediated by the photoperiod, by a phenotyping experiment. The measurement of the relative chlorophyll content under two different day length scenarios (8h & 16h) indicated a significant difference in barley ($p_{8h\ to\ 16h} < 0.001$). At the same time, no variation was observed between the tomato groups ($p_{8h\ to\ 16h} = 0.504$) (Fig 5A). The average relative chlorophyll content in leaves was 25.53 in barley genotypes in the 8h scenario and 35.95 in 16h. Analogously, the tomato values were 30.13 (8h) and 31.37 (16h). As these numbers already indicate, in a direct comparison of tomato and barley in the same day length scenario, tomato plants performed significantly better under 8h conditions ($p_{8h-tomato-barley} < 0.0003$). In contrast, the opposite was true in the 16h condition ($p_{16h-tomato-barley} < 0.04$).

## Conserved protein sequences and transcription patterns

Five hundred nineteen orthologous genes were identified based on their protein structure homology. These were used to estimate the magnitude of conservation in the expression level (Fig 6A, S7 Table). This 1-to-1 sequence similarity relationship indicates that these respective barley and tomato genes were more closely related than any other genes.

A principal component clustering of these genes exposed the higher transcriptional relation tissue-wise in this set of orthologues (Fig 6A). Compared to the collection of GO expression of all genes in Fig 3A, the tissues show increased transcriptional conservation in the group of

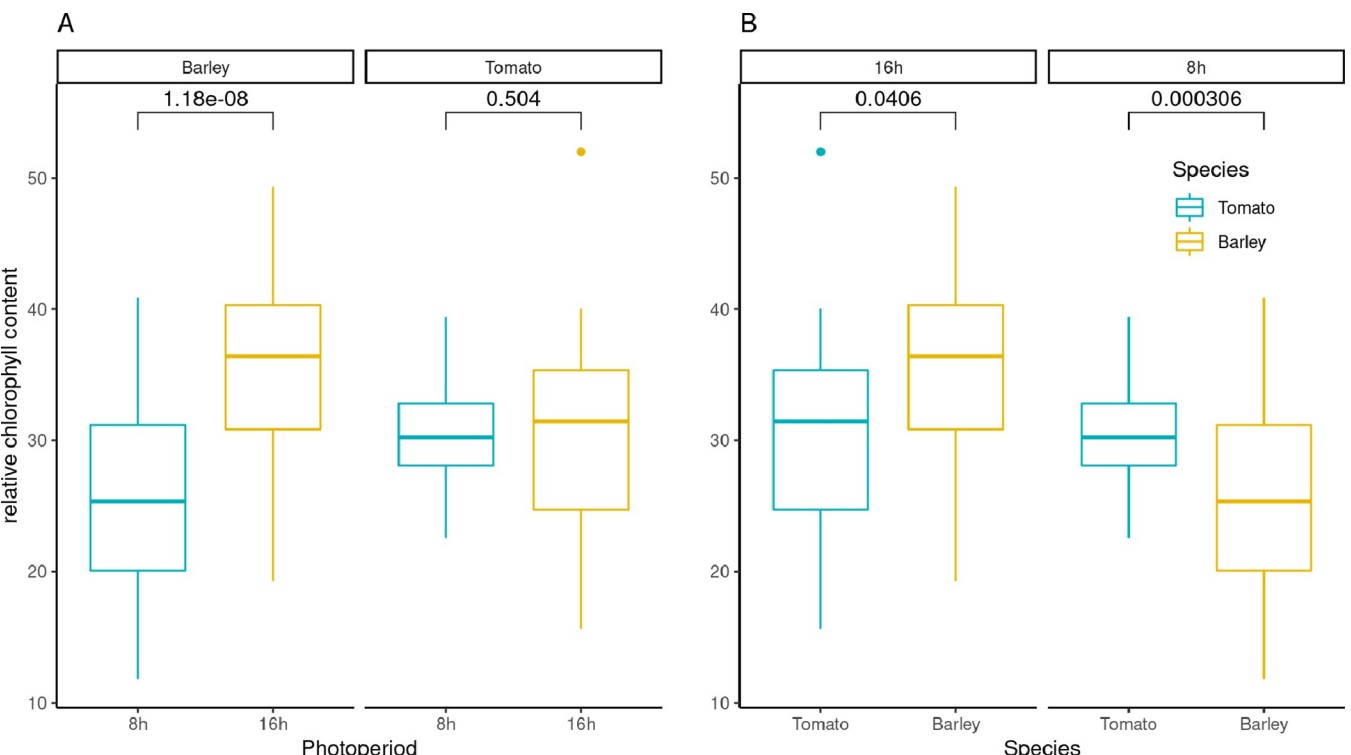

**Fig 5. Relative chlorophyll content assessed by an multispeQ gadget for ten replicated seedlings of barley (cultivar Scarlett) and tomato (cultivar Moneymaker) in two different photoperiods (8h, 16h day light).** Values above boxplots indicate the result of a t-test comparison.

orthologous genes. While the first component separates the root tissue, the shoot tissues are separated by the second component. The clustering of these orthologues in DEG and EEG results in two unequally sized groups. Based on a threshold of at least five reads in either set, 94 and 91 DEG were identified, comparing the species tissue-wise for root and shoot, respectively. The majority of these genes were not or only marginally expressed in barley. With a minimal expression threshold of five reads in both species, the number of DEG was reduced to 12 and 14 in root and shoot, respectively (Fig 6B). Six of these genes were found in both tissues. These include endopeptidase activity, proteolysis, structural constituent of ribosome, SNAP receptor activity, and response to stress and oxidation-reduction. These genes have a significant overexpression in the tomato tissues in common, with an average LFC value of ten.

The other EEG group was almost four times bigger (Fig 6B). Forty-one EEGs were identified in the root gene expression, while forty-five were detected in the shoot comparison. Seven of these were found in both tissues, including transcription coactivator activity, clathrin binding, hydrolase activity, protein binding, metabolic processes, and two endonuclease activity genes. The average expression of these genes was 298 normalized fragments, indicating an overall high expression level. Four were annotated as transcription factors for the EEG in the root, ten were related to protein binding, 14 were identified as enzymes, two were related to oxidation reactions, and four were identified as endonuclease enzymes. In shoot-related EEG observed genes, 14 enzymes were identified. Nine were described as transcription factors, five protein-binding-related genes, seven genes with oxidation background, and three endopeptidases. Several genes were annotated to more than one function.

High expressional conservation between the species was observed when clustered to biological processes, cellular components, and molecular function (Fig 5B). Especially the count of

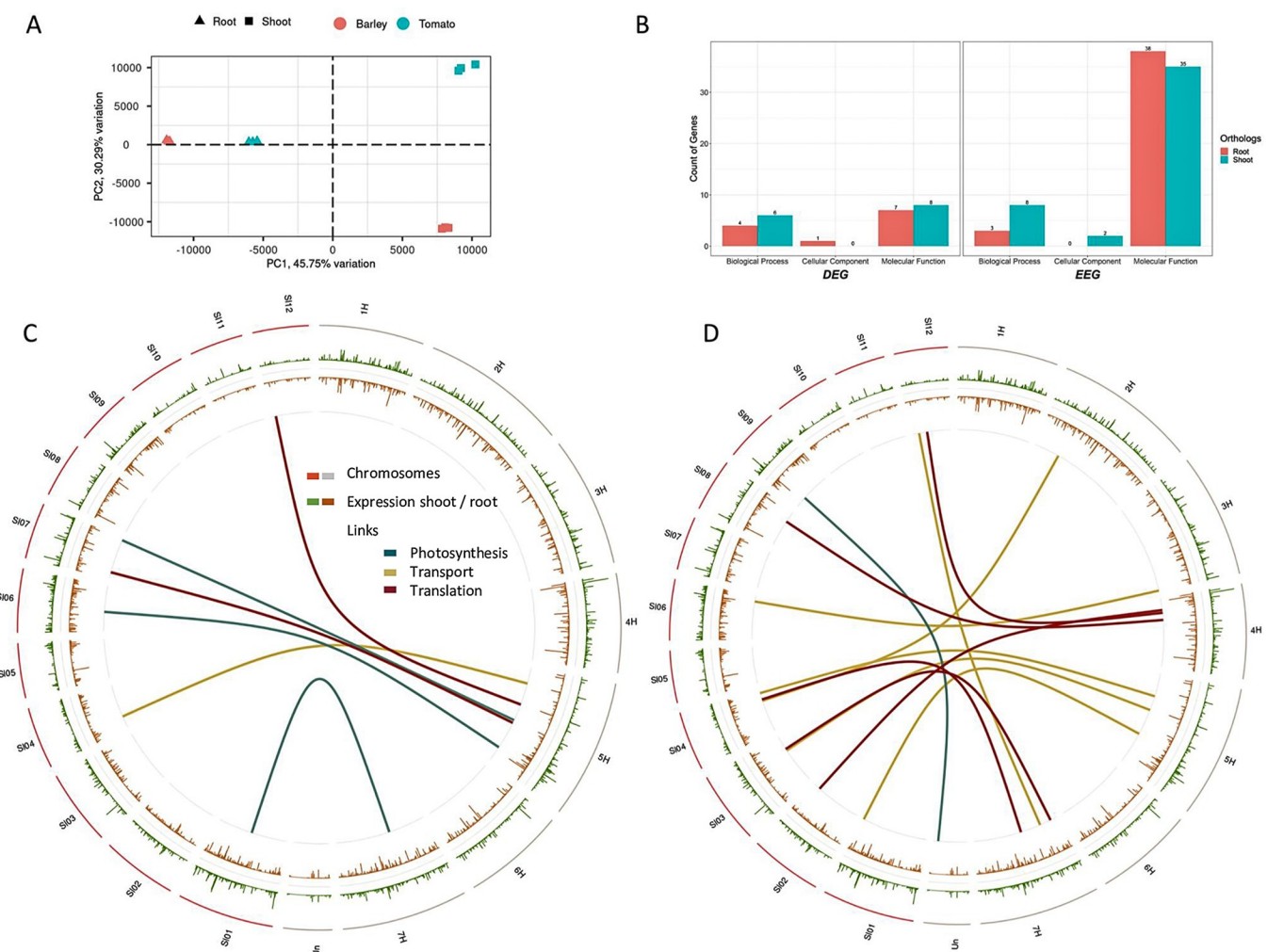

**Fig 6. Orthologous gene comparison between tomato and barley tissues.** A–principal component analysis of the orthologue genes, showing all three replicates for each tissue–species combination. B–Classification of differentially expressed genes (DEG) and equally expressed genes (EEG) in the three groups biological process, cellular component, and molecular function. The count of genes of each group is illuminated. C–Circos plot of the normalized expression of shoot (green) and root, meristems (brown) for tomato and barley, separated by chromosomes. Differentially expressed genes between barley and tomato for the three groups photosynthesis (dark green), transport (gold), and translation (dark red) are linked with lines in the center. D–similar to C, but showing the links between equally expressed genes.

EEG in the molecular function group is 4.3 and 5.4 times higher than the DEG for shoot and root, respectively. The other two groups show lower representation in both groups, which results in no variation between DEG and EEG.

Ultimately, we mapped the orthologous genes and compared their genomic position between the species. As the three groups transport, translation, and photosynthesis have caught the most interest, we compared the genomic loci between tomato and barley for DEG (Fig 6C), and EEG (Fig 6D) expressed orthologous genes. Three photosynthesis, one transport, and two translation-related genes were identified among the DEG group. Analog, one photosynthesis, six transport, and five translation-related orthologous genes were observed for the EEG group. The comparison of the genomic loci revealed a hotspot on barley's chromosome 5H, where five of six DEG orthologous genes were located. Besides, three EEG transport-related genes were found on 5H. Contrasting to barley, the distribution of the DEG and EEG orthologous genes did not indicate any clustering in the tomato genome.

In conclusion, the orthologues' significant variation in expression is related to a void barley root and shoot gene expression at the seedling stage. For the set of genes showing expression in both tissues, the majority is described by the group of molecular functions. Most of these genes indicated a similar expression.

## 4. Discussion

The functional analysis of these two divergent species was performed after a cultivation period of ten days, described by a photoperiod of 8h. While modern tomato varieties are day-length neutral [43], barley, as facultative long-day species [44], depends on long photoperiods to flower. Regarding this habit, short day-length was also observed to suppress growth and development in barley at the seedling stage [45]. Although the photoperiodic habit of species is generally determined, the day length type depends on the presents or expression level of specific candidate genes and, therefore, is interchangeable [43]. Furthermore, mutation breeding has created barley varieties with day-length neutral habits, indicating the potential to change the photoperiodic sensitivity [44].

This study aimed to identify significant variations and similarities of physiological development in early seedling stages between a day-length neutral tomato variety and a long-day barley variety. Therefore, root apices, comprising root meristem and root elongation zone of barley and tomato, were precisely harvested under a dissecting microscope. Likewise, shoot meristems comprising two emerging leaf primordia were gathered in both species. To homogenize the sampling process, we cut exactly 50 roots and shoot apices (as technical replicates) and pooled them in each biological replicate. The primary reason behind this sampling strategy was to target development-related genes and ensure the reproducibility of transcript data. Our data showed very highly similar gene expression among the individual biological replicates in each tissue in both species, suggesting that the adopted sampling strategy was appropriate (S2 & S3 Figs).

We used massive cDNA Ends (MACE) analysis instead of whole transcriptome sequencing by standard RNAseq approaches. MACE was preferred over RNAseq for two reasons. First, the bias of gene length should be avoided. While the gene length variation does not matter much in intraspecies transcriptome analysis, the comparison between species results in gene length expression bias. Both the orthologue genes and the gene ontology comparison on expression level could be biased by the gene length-related expression. Although the expression could have been corrected by the gene length, only sequencing the 3' single end of the gene gives higher confidence in the processed approach. Second, PCR duplicates are reduced due to the applied TrueQuant approach while sequencing. This should ensure minimal PCR bias during transcriptome sequencing. Ideally, each template molecule can be identified by its unique TrueQuant adapter sequence. Based on this, PCR copies can be determined and eliminated from the dataset, and uneven amplification and artifacts generated during the PCR amplification can be eliminated. Nevertheless, MACE also has some relevant disadvantages, like the comparably short read length and the sequencing on the 3' end. These two might have caused the significant loss of fragments throughout the alignment, filtering, and annotation process. As previously reported, inaccurate gene annotations might result in failed read annotations for those fragments with a start position beyond the annotated gene start [46]. This becomes a more relevant issue the shorter the reads are. Besides, the precise alignment of very short single-end reads is challenging. Maybe this was the other reason for the high number of unaligned reads.

In the functional analysis, the intraspecies comparison revealed a generally lower expression of genes in barley tissues than tomato. While only about 48% of all genes were expressed

in barley, almost 73% of all genes have shown evidence of activity in tomato. The higher number of expressed genes is also represented by a higher relative number of DEG in Tomato compared to barley (Fig 2). This is the first indication of lower differentiation activity in barley tissues. A GO enrichment, based on the occurrences of DEG, revealed twelve more variant terms in the tomato root to shoot comparison than for the barley tissues (Fig 3). Similarities were found for the terms cell wall organization, metabolic processes, and transcription factor binding. In addition, variations were observed for stress response and photosynthetic activity, which showed variance in the tomato tissues, but not between the barley tissues.

Based on an expression value comparison of root to shoot tissues, particular GO variations between the species were observed. Barley root tissue-specific ontologies were nutrient transportation, respirational aspects, and oxidative stress response. In tomato, gene overexpression in roots was found for nutrient uptake, transport, and storage groups. Additionally, elongational processes can be observed, indicating growth processes in the root tissue. Comparing the shoot level, both species have an overexpression in oxidative response classes and the photosystem I activity in common. Besides these, six additional GO classes related to the photosynthetic activity are overexpressed in the tomato shoot tissue. Overexpression of photosynthetic genes in the shoot is not unexpected, but the missing overexpression in barley tissue makes it remarkable. We found two gene copies of the Photosystem II reaction center W protein in tomato and one in barley. The high expression of the genes in tomato indicated pronounced photosynthetic activity. Contrasting, the same gene in barley was only marginally expressed in the shoot meristem.

Based on these Photosystem II transcript patterns of barley and tomato shoot tissues, we performed a small phenotypic experiment, where we measured the leaf chlorophyll content of barley and tomato seedlings, treated with 8h and 16h day-length. Based on the observations made in transcriptome comparison, We hypothesized that (I) tomato plants grown under 8h and 16h day length should not differ in their relative chlorophyll content–as it is a day-length neutral species; (II) contrasting to tomato, there is a difference between barley plants grown in 8h and 16h environments, and finally (III) the relative chlorophyll content of the leaves will be higher in tomato than barley under short-day conditions. This experiment validated that the barley genotype Scarlett was affected in its relative leaf chlorophyll content by the day length. A 30% reduction in the relative chlorophyll content was observed in the 8h compared to the 16h environment. Contrasting, the tomato genotype moneymaker did not facilitate any reduction in the chlorophyll content by the reduced day length. We concluded that all three hypotheses are fulfilled. Therefore, we consider the observed gene expression patterns to indicate reduced photosynthetic activity in the barley genotype Scarlett under 8h light regimes.

Furthermore, developmental gene ontologies and energy transforming processes indicated that tomato shoot tissue was associated with superior energy production under short-term light conditions compared to barley shoot tissue.

One might speculate this variation might also be caused by the tomato's lower seed weight than barley. This might force tomato seedlings to overweight the energy production and photosynthesis gene expression compared to barley. Overall, there were 2.7 times more gene ontologies found in tomato to vary between root and shoot tissues. This might indicate higher tissue specificity, probably associated with a more pronounced developmental variation under the given light regime.

The following comparison level supported the GO variations' higher physiological and metabolic activity in tomato tissue. Therefore, the LFC variation between root and shoot tissues on the intraspecies level was compared between the species. Two photosynthesis-related GO terms were highly expressed in the tomato shoot (Fig 2), but not in the barley shoot (Fig 1). Especially interesting is the low expression level of photosystem II-related genes in barley

shoot tissue. These two photosystems cover a different range of light absorbance (680nm, 720nm), which leads to a reduced energy transformation from less energetic light due to the reduced activity of the photosystem II [47].

Furthermore, the activity of photosystem II is reported to require more light than photosystem I [48]. Evolutionary variant extrinsic proteins might have a crucial effect on the structure and function of the photosystem II [49]. The reduced photosynthetic activity is framed in an overall reduced physiological activity in barley tissues. The reduced physiological activity was assumed based on the structural constituent of ribosomes, protein binding, NADPH dehydrogenase, or proton-transporting ATPase activity. The structural constituent of ribosomes plays a crucial role in regulating gene expression [50–52]. The count of protein-binding related genes has been observed to be one of the most expressed groups, which might be related to tissue differentiation processes [53] or the regulation of plant developmental processes by protein-protein interactions [54]. NADPH dehydrogenase, relevant in the respiratory chain [55,56], was more active in tomato tissues. The activity of NADPH dehydrogenases was reported to be dependent on $Ca^{2+}$ [57]. Calcium has several roles in plant development [58], and we observed the calcium ion binding expression also being higher in tomato tissues. This might indicate that a reduced calcium ion binding results in reduced activity of the NADPH dehydrogenases and, ultimately, reduced development and differentiation in barley tissues. The PCA shows a higher tissue distinction in tomato (Fig 3A), indicating a more pronounced developmental variation under the given light regime. The observed higher expression of stress-responsive genes in tomato tissues might be associated with unfavorable photoperiodic cycle conditions induced injuries, as described by Hillman (1956) [59].

The candidate genes *Cry1* and *Cry2* were selected as target genes, as these were described to have a relevant impact on tissue differentiation. Furthermore, the expression pattern was observed to change over time [7]. From the observation made in our experiment, one could conclude that tomato promotes shoot over root development (Fig 4). Contrasting, barley promotes the root growth overshoot development. But, as our experiment lacked a barley expression profile in a long-day light regime, we compared the observed *Cry1* and *Cry2* expression patterns to literature-obtained expression data. By this comparison, we aimed to answer whether the lower expression of *Cry1* in barley was associated with the species or the 8h light regime. Compared to the expression profile of Morex seedlings, derived by [60], *Cry1* was 2.9 times higher expressed in Morex under 16h light regime (external source) than Scarlett under 8h (S8 Fig). Similarly, *Cry2* was 2.14 times higher expressed in Morex compared to Scarlett. As these are two different genotypes and the sampling time point marginally differs, observed variations could be due to genotypic or time variations. Nevertheless, the expression variation between the environments for barley was highly significant, and the expression of Morex 16h was even beyond the level observed in tomato. Therefore, it could indicate that *Cry1* and *Cry2* alleles in Scarlett would be higher expressed than tomato orthologous under a 16h light regime.

Additionally, the expression conservation level between these two species was investigated by comparing orthologous genes. The hypothesis of the structural relation of genes leading to a higher level of equal expression can be confirmed. In the group of molecular functions, five times more genes showed equivalent expression compared to differential expression (Fig 5). The other two groups were shallowly covered, indicating that functional conservation beyond species levels is more likely in basic molecular functions. This statement is supported by the genes found to be EEG related to core functionalities, like enzyme activity and translational processes. All DEG genes have shown overexpression in tomato tissue, while most of these genes were not expressed in barley. This might change with a different light regime and might be another indicator of delayed and reduced physiological activity under short daylight.

Comparing the physical positions of orthologous genes annotated to photosynthesis, transport, and translation-related functions did not reveal that orthologous genes are not clustered on chromosomes. Similar results were observed for EEG, indicating that these two species are different in their genomic construction.

## 5. Conclusion

Applying a short photoperiod regime resulted in gene expression variations between tissues and species. The photoperiod is a relevant regulator for photosynthetic activity physiological and morphological differentiation for photoperiod-sensitive species like barley. As early sowing dates of spring-type crops result from climate change, the reduced growth under shorter day length conditions can indicate undesired lower productivity in unadapted varieties, especially with high latitudes. Breeding of new, less photoperiod-sensitive barley varieties might overcome delayed development and differentiation. Growth suspension by short photoperiod could benefit from genetic adjustments to avoid the coincidence of flowering and spring drought. This could retain high yields in rainfed crops through drought avoidance strategies by early root development.

## Supporting information

**S1 Fig. Workflow of the gene expression and functional analysis on three levels.** Each level is framed by a yellow, orange, or green square.
(TIF)

**S2 Fig.** The number of reads (in millions), mapped reads, and mapped in genes in Barley (A) and tomato gene annotation (B).
(TIF)

**S3 Fig.** Correlations among three biological replicates of root apices (BRx / TRx) and shoot apices (BSx/ TSx) in Barley (A) and tomato (B). Correlation is illustrated by color, shape, and additionally as a numerical value.
(TIF)

**S4 Fig. Gene ontology clustering results by applying AgriGo V2–51 significant GO terms have been identified between the root and shoot meristem in barley.** A–biological process; B–molecular function; C—cellular component.
(TIFF)

**S5 Fig. Gene ontology clustering results by applying AgriGo V2–63 significant GO terms have been identified between the root and shoot meristem in tomato.** A–biological process; B–molecular function; C—cellular component.
(TIFF)

**S6 Fig. Boxplot of the log10 transformed expression values for selected GO terms.** Each dot represents the expression level of a single gene, while the boxplot summarizes these single points. Root and shoot are separated by color, while the species are spatially divided. Two statistical tests were performed, between the tissues and between the species for each GO term individually. The results of the tissue-wise comparison are printed below the GO term name between the green and purple boxplots, while the species-related comparison is placed between the spatially separated boxplots.
(TIF)

**S7 Fig. Boxplot of the log10 transformed expression values for selected GO terms.** Each dot represents the expression level of a single gene, while the boxplot summarizes these single points. Root and shoot are separated by color, while the species are spatially divided. Two statistical tests were performed, between the tissues and between the species for each GO term individually. The results of the tissue-wise comparison are printed below the GO term name between the green and purple boxplots, while the species-related comparison is placed between the spatially separated boxplots. The GO term and function are illustrated above each boxplot. (TIF)

**S8 Fig. Normalized expression (RPM) of photomorphogenesis controlling genes *Cry1* and *Cry2*.** The expression is compared between tissues (x-axis) and species (color). Expression data for Morex, published by Liu et al. 2020 was added to compare barley expression patterns under short-day (Scarlett– 8h) and long day (Morex– 16h) photoperiods. (TIF)

**S1 Table. Barley gene expression of normalized expression value.** Basemean–mean expression over all six replicates, BasemeanA–root mean value; BasemeanB–shoot mean value. PAdj–Bonferroni adjusted p-value. BR_Rx–normalized expression values for each replicate separated BR = barley root. BSM_Rx–normalized expression values for each replicate of the barley shoots separately. (XLSX)

**S2 Table. Result of AgriGo V2 clustering of significantly expressed genes in barley root and shoot.** (XLSX)

**S3 Table. Barley gene ontology clustering based on expression values for each GO term.** Genecount–count of genes found for the GO term (identical between root and shoot); Avgr–Average expression value root; SDr–standard deviation for root tissue; MINr / MAXr–minimum and maximum expression values. Same presented for shoot tissue by Avgs, SDs, MINs and MAXs. Pval–probability value calculated by a generalized linear model (binomial distribution). Logfc–Logfoldchange value. FDR–adjusted p-value. (XLSX)

**S4 Table. Tomato gene expression of normalized expression value.** Basemean–mean expression over all six replicates, BasemeanA–root mean value; BasemeanB–shoot mean value. PAdj–Bonferroni adjusted p-value. TR_Rx–normalized expression values for each replicate separated TR = tomato root. TSM_Rx–normalized expression values for each replicate of the tomato shoot separately. (XLSX)

**S5 Table. Result of AgriGo V2 clustering of significantly expressed genes in tomato root and shoot.** (XLSX)

**S6 Table. Tomato gene ontology clustering based on expression values for each GO term.** Genecount–count of genes found for the GO term (identical between root and shoot); Avgr–Average expression value root; SDr–standard deviation for root tissue; MINr / MAXr–minimum and maximum expression values. Same presented for shoot tissue by Avgs, SDs, MINs and MAXs. Pval–probability value calculated by a generalized linear model (binomial distribution). Logfc–Logfoldchange value. FDR–adjusted p-value. (XLSX)

**S7 Table. Orthologous genes (1 to1 relationship) identified by OrthoMCL analysis of protein sequences between barley and tomato.**
(XLSX)

## Acknowledgments

We are grateful to Mrs. A. Bungartz for her help in sample preparation. Special thanks to Mrs. Anna Vlasova for her valuable suggestion in data analysis and to Mr. Md. Kamruzzaman for reading the manuscript.

## Author Contributions

**Conceptualization:** Heiko Schoof, Jens Léon, Ali Ahmad Naz.

**Data curation:** Michael Schneider.

**Formal analysis:** Michael Schneider, Lucia Vedder, Boby Mathew.

**Funding acquisition:** Jens Léon, Ali Ahmad Naz.

**Investigation:** Michael Schneider, Benedict Chijioke Oyiga, Ali Ahmad Naz.

**Methodology:** Lucia Vedder, Heiko Schoof, Ali Ahmad Naz.

**Project administration:** Jens Léon, Ali Ahmad Naz.

**Resources:** Benedict Chijioke Oyiga, Jens Léon.

**Software:** Boby Mathew.

**Supervision:** Jens Léon, Ali Ahmad Naz.

**Validation:** Heiko Schoof.

**Visualization:** Michael Schneider.

**Writing – original draft:** Michael Schneider, Lucia Vedder.

**Writing – review & editing:** Lucia Vedder, Benedict Chijioke Oyiga, Boby Mathew, Heiko Schoof, Jens Léon, Ali Ahmad Naz.

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
