## [Decision Letter · Decision Letter 0]

16 May 2022

PONE-D-22-07169Transcriptome profiling of barley and tomato shoot and root meristems unravels physiological variations underlying photoperiodic sensitivityPLOS ONE

Dear Dr. Schneider,

Thank you for submitting your manuscript to PLOS ONE. After careful consideration, we feel that it has merit but does not fully meet PLOS ONE’s publication criteria as it currently stands. Therefore, we invite you to submit a revised version of the manuscript that addresses the points raised during the review process.

We look forward to receiving your revised manuscript.

Kind regards,

Raffaella Balestrini

Academic Editor

PLOS ONE

Journal Requirements:

Reviewers' comments:

Reviewer's Responses to Questions

**Comments to the Author**

1. Is the manuscript technically sound, and do the data support the conclusions?

Reviewer #1: Partly

2. Has the statistical analysis been performed appropriately and rigorously? 

Reviewer #1: Yes

3. Have the authors made all data underlying the findings in their manuscript fully available?

Reviewer #1: Yes

4. Is the manuscript presented in an intelligible fashion and written in standard English?

Reviewer #1: Yes

5. Review Comments to the Author

Reviewer #1: The manuscript by Schneider et al. deals with global transcriptome profiling of plant development genes in the seedling stage of root and shoot apical meristems of a photoperiod-sensitive species (barley) and a photoperiod insensitive species (tomato) in short-day conditions (8 h)

The application of a short photoperiod regime resulted in gene expression variations between tissues and species. In particular, in the functional analysis, the intraspecies comparison revealed a generally lower expression of genes in barley tissues than tomato.

The research could be interesting for the agricultural field in the face of the climate change, however the manuscript appears to be too descriptive due to the lack of multi-disciplinarity. The results obtained with the MACE approach and subsequent bioinformatic analyses are quite interesting, however no other methods were used to substantiate these data. I would expect some morphological and biochemical data along with transcriptomics. The effects due to short-day conditions should be visible also at morphological and biochemical level.

Moreover, I think that a validation of the MACE analysis should be carried out, maybe by analyzing the expression of some identified and selected genes by qPCR.

Finally, I could not find example of specific genes differentially expressed in GO categories. Only a general description is provided. The authors should also report some specific gene name and number.

Fig 2 D. I’m surprised to read “vasoactive intestinal polypeptide receptor activity”. Can the authors give an explanation?

Materials and Methods

Authors should report some additional details about the humidity level in the growth chamber. This parameter is missing

The manuscript contains a number of typographical errors that will require to be corrected. Authors should check carefully the text and correct all the English mistakes. I would also use the term orthologous genes instead than ortholog genes.

6. PLOS authors have the option to publish the peer review history of their article (what does this mean?). If published, this will include your full peer review and any attached files.

Reviewer #1: No

---

## [Author Response · Author response to Decision Letter 0]

13 Jul 2022

>>>>The research could be interesting for the agricultural field in the face of the climate change, however the manuscript appears to be too descriptive due to the lack of multi-disciplinarity. The results obtained with the MACE approach and subsequent bioinformatic analyses are quite interesting, however no other methods were used to substantiate these data. I would expect some morphological and biochemical data along with transcriptomics. The effects due to short-day conditions should be visible also at morphological and biochemical level. 

A: We thank the reviewer for this suggestion. To accommodate this idea, we additionally performed an experiment on the photosystem II activity und short (8h) and long day (16h) conditions in a growth chamber. The settings were described in the manuscript, l.155-164. As previously reported in the manuscript, we observed a variant expression of the photosystem II in barley and tomato shoot tissues. Therefore, we set up the hypothesis that such variations in expression should result in measurable differences in the relative chlorophyll content of the leaves. Furthermore, we hypothesized that the relative chlorophyll content should 

• not be different between tomato plants grown under 8h and 16h daylength conditions

• should be different between barley plants grown under 8h and 16h daylength conditions

• be higher in tomato compared to barley leaves under 8h conditions.

We tested these hypotheses by measuring the relative chlorophyll content in the oldest and youngest leaf to the end of the daylight phase and found evidence supporting all our hypotheses (line 317-325, line 429-448).

>>>>Moreover, I think that a validation of the MACE analysis should be carried out, maybe by analyzing the expression of some identified and selected genes by qPCR.

A: We had considered performing qPCR studies to validate some of our gene-expression findings. But based on the fact that (I) all three biological replicates showed a striking high expression correlation and (II) previous studies had shown close correlations between qPCR and RNAseq data (1–5), we decided to eschew those experiments.

>>>>Finally, I could not find example of specific genes differentially expressed in GO categories. Only a general description is provided. The authors should also report some specific gene name and number.

A: Thank you for this suggestion. We included for some selected GO categories the number of genes, name, and function of selected genes (line 237pp, line 245pp, line 283pp). If a reader is more interested in finding information on all gene expressions and the responding category (GO), a reader can have a look at the supplementary tables. We tried to improve the readability of these tables as well. We hope the performed changes are in line with your expectations.

>>>>Fig 2 D. I'm surprised to read "vasoactive intestinal polypeptide receptor activity". Can the authors give an explanation?

A: We thank the reviewer for this valuable comment. We dissected our data and found six genes associated with this gene ontology (GO:0004999). While the average normalized gene expression was 364 reads in the tomato root meristem across all those six genes, on average 4 reads were examined for the shoot meristem. One additional important fact we realized when dissecting the data was that all six genes were also associated with the DNA binding GO term GO:0003677. As the vasoactive intestinal polypeptide receptor activity seems to you and to us to be an unlikely candidate class, we changed the naming in the figure accordingly to DNA binding. We thank you for mentioning this aspect and for improving our manuscript with this comment.

>>>>Authors should report some additional details about the humidity level in the growth chamber. This parameter is missing

A: We added this detail in line 80. 

>>>>The manuscript contains a number of typographical errors that will require to be corrected. Authors should check carefully the text and correct all the English mistakes. I would also use the term orthologous genes instead than ortholog genes.

A: We revised the manuscript another time to rule out language-related errors.

As you may notice, we have tried our best to address the concerns of reviewer 1. We are grateful to the reviewers and the editor for the time, efforts, and valuable suggestions that resulted in a better manuscript. 

References:

1. Shi Y, He M. Differential gene expression identified by RNA-Seq and qPCR in two sizes of pearl oyster (Pinctada fucata). Gene. 2014 Apr 1;538(2):313–22. 

2. Wu AR, Neff NF, Kalisky T, Dalerba P, Treutlein B, Rothenberg ME, et al. Quantitative assessment of single-cell RNA-sequencing methods. Nat Methods 2013 111 [Internet]. 2013 Oct 20 [cited 2022 Jun 17];11(1):41–6. Available from: https://www.nature.com/articles/nmeth.2694

3. Asmann YW, Klee EW, Thompson EA, Perez EA, Middha S, Oberg AL, et al. 3' tag digital gene expression profiling of human brain and universal reference RNA using Illumina Genome Analyzer. BMC Genomics [Internet]. 2009 Nov 16 [cited 2022 Jun 17];10(1):531. Available from: https://bmcgenomics.biomedcentral.com/articles/10.1186/1471-2164-10-531

4. Griffith M, Griffith OL, Mwenifumbo J, Goya R, Morrissy AS, Morin RD, et al. Alternative expression analysis by RNA sequencing. Nat Methods 2010 710 [Internet]. 2010 Sep 12 [cited 2022 Jun 17];7(10):843–7. Available from: https://www.nature.com/articles/nmeth.1503

5. Everaert C, Luypaert M, Maag JLV, Cheng QX, DInger ME, Hellemans J, et al. Benchmarking of RNA-sequencing analysis workflows using whole-transcriptome RT-qPCR expression data. Sci Reports 2017 71 [Internet]. 2017 May 8 [cited 2022 Jun 17];7(1):1–11. Available from: https://www.nature.com/articles/s41598-017-01617-3

---

## [Decision Letter · Decision Letter 1]

18 Aug 2022

Transcriptome profiling of barley and tomato shoot and root meristems unravels physiological variations underlying photoperiodic sensitivity

PONE-D-22-07169R1

Dear Dr. Schneider,

We’re pleased to inform you that your manuscript has been judged scientifically suitable for publication and will be formally accepted for publication once it meets all outstanding technical requirements.

Kind regards,

Raffaella Balestrini

Academic Editor

PLOS ONE

Additional Editor Comments (optional):

Reviewers' comments:

Reviewer's Responses to Questions

**Comments to the Author**

1. If the authors have adequately addressed your comments raised in a previous round of review and you feel that this manuscript is now acceptable for publication, you may indicate that here to bypass the “Comments to the Author” section, enter your conflict of interest statement in the “Confidential to Editor” section, and submit your "Accept" recommendation.

Reviewer #1: All comments have been addressed

2. Is the manuscript technically sound, and do the data support the conclusions?

Reviewer #1: Yes

3. Has the statistical analysis been performed appropriately and rigorously? 

Reviewer #1: Yes

4. Have the authors made all data underlying the findings in their manuscript fully available?

Reviewer #1: Yes

5. Is the manuscript presented in an intelligible fashion and written in standard English?

Reviewer #1: Yes

6. Review Comments to the Author

Reviewer #1: The manuscript has been improved and the authors made all the corrections/modifications according to my suggestions.

7. PLOS authors have the option to publish the peer review history of their article (what does this mean?). If published, this will include your full peer review and any attached files.

Reviewer #1: No

---

## [Editor Report · Acceptance letter]

2 Sep 2022

PONE-D-22-07169R1 

Transcriptome profiling of barley and tomato shoot and root meristems unravels physiological variations underlying photoperiodic sensitivity 

Dear Dr. Schneider:

I'm pleased to inform you that your manuscript has been deemed suitable for publication in PLOS ONE. Congratulations! Your manuscript is now with our production department. 

Kind regards, 

on behalf of

Dr Raffaella Balestrini 

Academic Editor

PLOS ONE